# Inflammatory Mechanisms Underlying Nonalcoholic Steatohepatitis and the Transition to Hepatocellular Carcinoma

**DOI:** 10.3390/cancers13040730

**Published:** 2021-02-10

**Authors:** Moritz Peiseler, Frank Tacke

**Affiliations:** 1Department of Hepatology & Gastroenterology, Charité University Medicine Berlin, 13353 Berlin, Germany; moritz.peiseler@gmx.de; 2Snyder Institute for Chronic Diseases, Cumming School of Medicine, University of Calgary, Calgary, AB T2N 4N1, Canada; 3Department of Pharmacology & Physiology, University of Calgary, Calgary, AB T2N 4N1, Canada

**Keywords:** nonalcoholic steatohepatitis, hepatocellular carcinoma, cancer immunotherapy, inflammation, macrophages, innate immunity

## Abstract

**Simple Summary:**

Nonalcoholic fatty liver disease (NAFLD) is the most common chronic liver disease affecting a quarter of the global population and carries the risk of developing malignant liver cancer. Inflammation is considered paramount in the progression of the disease and many different immune cells and pathways have been implicated in the development of NAFLD. Novel techniques in basic immunology have substantially enhanced the possibilities to study inflammation and the heterogeneity of immune cells. Based on recent studies, we provide a review of novel paradigms emerging in steatohepatitis and the development of hepatocellular carcinoma and outline the multifaceted contributions of immunity to advancing NAFLD. End-stage NAFLD with liver cancer has an abysmal prognosis and effective medical therapies are lacking, therefore a better understanding of the disease mechanisms will ultimately help to improve patients’ care.

**Abstract:**

Nonalcoholic fatty liver disease (NAFLD) is a rising chronic liver disease and comprises a spectrum from simple steatosis to nonalcoholic steatohepatitis (NASH) to end-stage cirrhosis and risk of hepatocellular carcinoma (HCC). The pathogenesis of NAFLD is multifactorial, but inflammation is considered the key element of disease progression. The liver harbors an abundance of resident immune cells, that in concert with recruited immune cells, orchestrate steatohepatitis. While inflammatory processes drive fibrosis and disease progression in NASH, fueling the ground for HCC development, immunity also exerts antitumor activities. Furthermore, immunotherapy is a promising new treatment of HCC, warranting a more detailed understanding of inflammatory mechanisms underlying the progression of NASH and transition to HCC. Novel methodologies such as single-cell sequencing, genetic fate mapping, and intravital microscopy have unraveled complex mechanisms behind immune-mediated liver injury. In this review, we highlight some of the emerging paradigms, including macrophage heterogeneity, contributions of nonclassical immune cells, the role of the adaptive immune system, interorgan crosstalk with adipose tissue and gut microbiota. Furthermore, we summarize recent advances in preclinical and clinical studies aimed at modulating the inflammatory cascade and discuss how these novel therapeutic avenues may help in preventing or combating NAFLD-associated HCC.

## 1. Introduction

Nonalcoholic fatty liver disease (NAFLD) is the most prevalent chronic liver disease, affecting about 25% of the global population and is the fastest growing cause of hepatocellular carcinoma (HCC) [1]. The rising prevalence of NAFLD is parallel to the global increase in obesity, metabolic syndrome and type 2 diabetes mellitus. Importantly, among NAFLD patients, 80% are obese, and in obesity, the prevalence of NAFLD increases to 60 –95% [2,3]. Given the high number of overweight children and adolescents globally, it is estimated that the burden of NAFLD will increase further [4,5]. 

NAFLD comprises a spectrum of chronic liver disease, ranging from isolated hepatic steatosis, to nonalcoholic steatohepatitis (NASH) characterized by steatosis, necroinflammation and hepatocyte injury with or without fibrosis, to cirrhosis and/or HCC [6,7]. In the United States, 15–25% of individuals with steatosis develop NASH, and among the group of NASH patients, 25% further progress to cirrhosis [8]. With NAFLD rising rapidly in recent years, NASH cirrhosis is now a leading cause of liver transplantation [9]. HCC is an additional sequela associated with NAFLD and the majority of NAFLD-related HCC develop in cirrhotic livers, however 20–30% develop in the absence of cirrhosis [10]. Hepatocellular carcinoma currently ranks sixth in global cancer incidence and is the fourth leading cause of cancer mortality [11]. A population based study in the US identified NAFLD as the most common etiological risk factor for HCC [12,13]. In summary, the rising global prevalence of NAFLD and HCC represent major clinical challenges, further complicated by the lack of approved drugs for NAFLD and organ shortage for transplantation. 

The pathogenesis of NAFLD and transition to HCC is multifactorial, yet inflammation is considered a key element of progression to advanced disease and HCC [14]. While our understanding of the inflammatory processes in NAFLD is evolving, many elements are still enigmatic. In this review, we summarize immune-mediated mechanisms of NASH and the transition to associated HCC. In a complex multisystem disease, such as NAFLD, the immune system is constantly torn between inflammation, repair and tumor surveillance—in light of novel immunotherapies for HCC, an improved understanding of the precise immune pathways is necessary. 

## 2. The Role of Inflammation in NAFLD

Inflammation in response to tissue injury without infection is termed “sterile inflammation” [15] and is characterized by an initial hyperinflammatory phase, clearing tissue debris, followed by a resolution phase of tissue healing and an anti-inflammatory environment [16]. The sterile inflammatory cascade has been optimized over millions of years of evolution and each step is necessary for healing [17,18,19,20]. However, in the state of over-nutrition and excessive lipid storage, as seen in obesity, a new type of sterile inflammation was identified, coined “metabolic inflammation”. Metabolic inflammation is characterized by low grade smoldering immune activation without adequate resolution, triggered by metabolic pathways, insulin resistance and lipotoxicity [21]. The earliest model for the pathogenesis of steatohepatitis was a “two-hit” hypothesis; lipid accumulation provides a ”first hit” to the liver, followed by a “second hit” revolving around oxidative stress, lipid peroxidation and necroinflammation precipitating NASH [22]. It is now apparent that NAFLD is a complex multifactorial disease, where a genetically susceptible host and environmental modifiers including diet, lifestyle and microbiota act in concert to modify adaptation to caloric excess [23]. Numerous single-nucleotide polymorphisms (SNPs) have been identified as genetic modifiers of NAFLD, with SNPs in the genes encoding PNPLA3 (platin-like phospholipase domain containing 3) and TM6SF2 (transmembrane 6 superfamily member 2) receiving the most attention [24]. Both PNPLA3 and TM6SF2 are involved in lipid metabolisms and not directly related to inflammation, however, gene polymorphisms in a number of inflammatory genes, including interleukin(IL)-1 and tumor necrosis factor (TNF) were identified in studies and are discussed elsewhere [25]. Sentinel cells in the liver sense excess metabolites, damaged hepatocytes and bacterial products and translate those signals into immune responses, resulting in steatohepatitis [26]. Inflammation in the context of fatty liver is not a one-way route towards progression, but rather a tug of war between necroinflammation and phases of resolution. Delineating the precise contributions of inflammatory pathways, cells and their temporal kinetics is warranted for developing novel therapies (Table 1).

### 2.1. Macrophages and Monocytes

The liver harbors the most abundant population of tissue macrophages in the human body, termed Kupffer cells (KC), which are embryonically derived and self-maintain locally [27,28] (Figure 1). As intravascular sentinels, Kupffer cells constantly scavenge the blood for particulate matter. In inflammatory settings, the pool of liver macrophages is augmented by a robust influx of bone marrow-derived monocytes that give rise to monocyte-derived macrophages (MoMF) [29]. Monocyte-derived macrophages are functionally and phenotypically distinct from Kupffer cells and infiltrate in response to inflammatory stimuli [30]. However, upon depletion of all Kupffer cells, there is remarkable plasticity between the two lineages [31,32]. 

In NAFLD, lipid excess leads to hepatocyte injury and the release of damage-associated molecular patterns (DAMPs) [33]. Kupffer cells detect DAMPs, triggering their activation and the release of proinflammatory mediators, representing an initiating step in steatohepatitis [29]. Animal studies showed that depletion of Kupffer cells attenuates NASH [34,35]. Recent studies have uncovered some of the mechanisms that lead to macrophage activation in steatohepatitis. Lipid-induced stimulation of hepatocytes triggered the release of extracellular vesicles containing tumor necrosis factor-related apoptosis-inducing ligand (TRAIL), which in turn activated macrophages to produce interleukin (IL)-1β and IL-6 [36]. Steatosis-induced downregulation of the glucocorticoid-induced leucine zipper was found to drive a proinflammatory phenotype in Kupffer cells [37]. Another hepatocyte-derived signal skewing Kupffer cells towards a proinflammatory phenotype in NASH was histidine-rich glycoprotein and mice lacking histidine-rich glycoprotein were protected from experimental steatohepatitis [38]. Metabolic disturbances, oxidative stress and translocated bacterial products were shown to activate Kupffer cells via Toll-like receptors (TLRs), particularly TLR4, in animal models of NASH, resulting in increased NF-κB signaling and proinflammatory cytokine production [39]. In a feed-forward-loop, metabolic stress sensed by Kupffer cells and subsequent release of proinflammatory cytokines induces the production of complement factors in hepatocytes, further amplifying the inflammatory cascade [26]. Among the plethora of receptors expressed by Kupffer cells are scavenger receptors such as CD36. In NASH patients, CD36 was upregulated and mechanistically, this increased cholesterol crystals in lysosomal compartments of Kupffer cells, leading to lysosomal damage and production of proinflammatory mediators [40]. 

A number of sophisticated studies have recently delineated the fate of embryonic Kupffer cells and infiltrating macrophages in experimental steatohepatitis using novel techniques including single-cell sequencing and genetic lineage tracing in mice [41,42,43] (Figure 1). In a set of experiments using bone marrow chimeras and parabiosis, Kupffer cell self-renewal was impaired and a higher rate of apoptotic Kupffer cells was observed, favoring engraftment of bone-marrow-derived Kupffer cells [41]. Furthermore, monocyte-derived Kupffer cells had a lower capability to store lipids compared to embryonic Kupffer cells and a more proinflammatory gene signature [41]. Interestingly, programmed cell death of Kupffer cells is a protective mechanism in *Listeria monocytogenes* infection [44], and in the context of chronic metabolic inflammation, this protective mechanism of initiating inflammation might be overturned. Another recent study used single-cell transcriptomics in mice fed a Western diet and similarly, identified a reduction in embryonic Kupffer cells and replacement with monocyte-derived macrophages [42]. This study identified additional subsets of liver macrophages in steatohepatitis, namely monocyte-derived Kupffer cells and a population termed lipid-associated macrophages, expressing osteopontin, with different gene expression profiles with regards to lipid metabolism and inflammation. Interestingly, the authors could not detect proinflammatory changes in embryonic Kupffer cells, suggesting many of the inflammatory changes found previously might be related to infiltrating macrophages [42]. This was in line with another recent study in obese humans and mice, concluding a proinflammatory reprogramming was not detectable in Kupffer cells [45]. Specialized subsets of liver macrophages have recently been identified in human cirrhosis and were subsequently termed scar-associated macrophages [46]. These subsets share markers such as TREM-2 and CD9, in line with another study investigating human and murine NASH, that found equivalent macrophage subsets [47]. Osteopontin was also identified as a biomarker in NASH patients [48]. Furthermore, blocking osteopontin in experimental NASH had protective effects [49,50,51]. Mechanistically, osteopontin induced collagen production in hepatic stellate cells, aggravating liver fibrosis in mice [52,53]. Another recent study investigated epigenetic changes in steatohepatitis in mice [43]. Congruent with the aforementioned studies, loss of embryonic Kupffer cells and replacement with different subsets of monocyte-derived Kupffer cells and macrophages was found in steatohepatitis, including a population expressing CD9 and TREM-2, that localized in the fibrotic niche, thus corresponding to scar-associated macrophages found in humans [43,46]. Furthermore, epigenetic reprogramming of liver X receptor (LXR), which conforms Kupffer cell identity, impaired Kupffer cell survival and promoted scar-associated macrophages [43]. In summary, these studies broaden our understanding of macrophage heterogeneity in NASH, identifying a conserved subset expressing TREM-2 and CD9, located in proximity to fibrosis. A caveat is that steatohepatitis in mouse models develops over weeks rather than years as in humans and is possible, that over a longer time course, the differences in genetic profiles in monocyte-derived cells eventually adopt to embryonic Kupffer cells [54]. Furthermore, a functional correlate of the different subsets has yet to be determined.

In mice, two subsets of monocytes are found in blood, proinflammatory monocytes, characterized by high expression of CC-chemokine receptor 2 (CCR2) and patrolling monocytes, defined by expression of the fractalkine receptor CX3CR1 [55]. In humans, monocytes are categorized as classical (CD14highCD16-), intermediate (CD14+CD16+) and non-classical (CD14-CD16high) monocytes [56]. Monocytes give rise to macrophages with a proinflammatory or a repair phenotype, depending on the (necessary) cues provided by the liver microenvironment [57], and furthermore, these cells can switch phenotype [58]. Proinflammatory monocytes are known drivers of steatohepatitis and accumulate mainly through the CCL2-CCR2-axis [59,60,61]. While CCR2 is expressed primarily by proinflammatory monocytes, the corresponding chemokine C-C motif ligand 2 (CCL2) is expressed by resident liver cells such as Kupffer cells, activated stellate cells or damaged hepatocytes [62]. Blocking CCL2 pharmacologically alleviated experimental NASH [63]. Furthermore, the therapeutic use of a CCR2/CCR5 antagonist reduced monocyte recruitment to the liver in models of steatohepatitis and thus reduced insulin resistance, NASH activity and fibrosis [64]. In patients with NASH, CCL2/CCR2 is upregulated and an accumulation of CCR2+ macrophages is found around the portal tracts, correlating with disease stage, suggesting similar mechanisms are involved [64,65]. Additional pathways for monocyte/macrophage recruitment in NASH are CXCR3-CXCL10 [66,67] and CCR9-CCL25 [68]. In summary, while proinflammatory monocytes/macrophages are known drivers of steatohepatitis in humans and mice, emerging evidence suggests the presence of a functional continuum of infiltrating macrophages, in part replenishing Kupffer cells, and more studies are necessary to comprehend to full scope of their role in NAFLD. 

### 2.2. Neutrophils

Neutrophil infiltration is a frequent histological observation in NAFLD and associated with disease progression [26]. In a rodent study, antibody-mediated neutrophil depletion halted steatohepatitis [69]. Mechanistically, neutrophil effector molecules such as proteases, elastase and myeloperoxidase were shown to be involved in liver damage [70]. In addition, a recent study demonstrated that mice deficient in neutrophil elastase, were protected from Western diet-induced NASH [71]. Neutrophil extracellular traps (NETs) are a recently identified killing mechanism of neutrophils and implicated in a number of inflammatory conditions [72,73,74]. A recent study identified a role for NETs in NAFLD [75]. The author found elevated biomarkers of NETs in the serum of patients with NASH, detected NET formation in murine NASH and blocking NETs resulted in attenuated steatohepatitis [75]. Of note, early neutrophil influx is absent in other models of NAFLD and it is thus unclear if neutrophils are involved in the early pathogenesis [76], however, once the liver parenchyma is inflamed, neutrophils certainly contribute and might cause additional damage, even though emerging evidence suggests that neutrophils also partake in tissue repair [17,18]. 

### 2.3. Dendritic Cells

Dendritic cells (DCs) are professional antigen-presenting cells, bridging innate and adaptive immunity [77]. In homeostasis, DCs are part of the tolerogenic environment in the liver, however upon perturbation, liver DCs become proinflammatory and secrete numerous cytokines [26]. DCs in the liver are divided into myeloid (or conventional) DCs (mDCs), expressing CD11c, and plasmacytoid DCs (pDCs), expressing CD123+ [78,79]. The role of DCs in NAFLD pathogenesis is unclear as contradictory evidence exists [80]. In a rodent study using the methionine-choline-deficient (MCD) diet, DCs were increased in the liver of mice, however, upon diphtheria toxin-induced depletion of CD11c+ cells, steatohepatitis exacerbated and DC ablation resulted in the expansion of CD8+ T cells and reduction in regulatory T cells, indicating a regulatory role [81]. Using this depletion approach carries the caveat that CD11c is expressed by other immune cells (which can include Kupffer cells and other macrophages in the liver). A recent study used a more targeted approach by depleting CD103+ DCs to show that mice deficient in CD103+ DCs had more severe steatohepatitis, confirming a protective role for CD103+ cDCs [82]. It was suggested that DC lipid content might influence their regulatory vs. inflammatory role, as DCs with high lipid content had a proinflammatory phenotype compared to low lipid DCs [83]. More recent evidence found CX3CR1-expressing myeloid DCs increased in mice treated with MCD diet, producing high levels of TNF and blocking CX3CR1 ameliorated steatohepatitis in mice, suggesting a disease promoting role for myeloid DCs in NAFLD [84,85]. These seemingly contradictory results likely represent the different subsets of DCs that were investigated, and more studies are needed to precisely characterize DCs in steatohepatitis at different stages. 

### 2.4. Natural Killer Cells

Natural killer (NK) cells are effector cells of the innate immune system with crucial functions in infection and tumor surveillance [86,87]. NK cells express CD56 and are classified as CD56^bright^ NK cells in tissues with the ability to release cytokines, whereas CD56^dim^ NK cells are preferentially found in blood and are characterized by their direct killing capacity [88,89]. NK cells were shown to have an important role in regulating liver fibrosis by directly killing activated hepatic stellate cells via the receptors NKG2D and NKp46 and the p38/PI3K/AKT pathway [90,91,92]. In patients with NAFLD, NK cells were increased in the liver [93]. In addition, a study found elevated expression of NKG2D on NK cells in patients with NASH [94]. Another recent study found decreased CD56^bright^ NK cells and increased CD56^dim^ NK cells in livers of NAFLD patients and interestingly, liver stiffness measurement negatively correlated with total NK cell numbers [95]. A study using the MCD diet in mice, showed a protective role for NK cells through the production of interferon (IFN)-γ [96]. Taken together, these studies suggest a protective role for NK cells in NAFLD and liver fibrosis.

### 2.5. Natural Killer T Cells 

Natural killer T (NKT) cells are innate-like lymphocytes that express markers of T cells and NK cells, regulate an array of immune responses and produce large quantities of Th1 or Th2 cytokines [97]. Recent studies showed the existence of at least two subsets: type I NKT (iNKT) cells are abundantly found in the murine liver, and type II NKT cells, that are more frequent in humans [98]. Since NKT cells are activated via lipid antigens, a role in NAFLD has been suggested [99]. Indeed, mice lacking NKT cells were shown to be more susceptible to developing fatty liver [100], and leptin-deficient mice displayed reduced hepatic steatosis after adoptive transfer of NKT cells [101]. Mechanistically, Kupffer cell derived IL-12 was shown to mediate early loss of NKT cells [102]. These studies suggest a regulatory role for NKT cells; however, other studies demonstrated opposing findings. Mice deficient in NKT cells given the MCD diet exhibited reduced fibrosis and in patients with advanced NASH cirrhosis, the authors noted a stark increase in NKT cells [103]. Mechanistically, mice deficient in NKT cells had reduced hedgehog pathway activation and reduced osteopontin expression compared to wild type mice and these factors stimulated stellate cells to become profibrotic [104]. In keeping with an immune modulating role, a recent study found that in experimental NASH in mice lacking iNKT cells, infiltration of neutrophils, CD8+ T cells and proinflammatory macrophages were diminished and stellate cell activation reduced [105]. Chemokine receptor CXCR6 and its ligand CXCL16 were identified as critical for recruitment of NKT cells to the liver and in two independent liver injury models, inhibiting CXCR6 ameliorated liver injury and fibrosis [106]. It is important to note, however, that NKT cell numbers and phenotypes vary tremendously between mice and humans, hampering the direct translation of findings from experimental mouse models to human disease. Possibly, additional innate lymphocyte populations such as mucosal associated invariant T cells (MAIT cells) in human liver exert some of the iNKT cell related functions identified in mice [27].

### 2.6. The Role of the Adaptive Immune System

Besides the well-established role of innate immunity in steatohepatitis [26,107], mounting evidence also suggests a role for adaptive immunity [108,109,110]. A frequent histological finding in NASH is diffuse lymphocytic infiltration, and lymphocytes are found in periportal infiltrates associated with NASH ductular reactions [111,112]. Furthermore, T and B cells were shown to form focal aggregates resembling ectopic lymphoid structures in a high number of patients with NASH [112,113]. Lastly, mice deficient in T cells failed to develop fructose-induced steatohepatitis, thus in principle demonstrating a role for T cells in NASH [114]. 

In mice and patients with steatohepatitis, CD8+ T cells were shown to accumulate in the liver, and their pharmacologic or genetic ablation ameliorated steatosis, insulin resistance and inflammation [115,116,117]. Mechanistically, type I interferon was the central pathway to govern liver CD8+ T cell infiltration in steatohepatitis [116]. In addition, numerous studies described CD4+ T cells in the livers of patients with NASH and in experimental models [110]. Infiltrating CD4+ T cells in mice fed a high fat diet were polarized towards Th1, expressing cytokines such as IFN-γ and TNF-α [118]. Furthermore, patients with NAFLD showed increased frequencies of IFN-γ producing CD4+ T cells in liver and blood [119,120] and mice deficient in IFN-γ developed less severe steatohepatitis and fibrosis in the MCD model [121]. A recent study demonstrated a role for α4β7-mediated recruitment of CD4+ T cells to the liver in mice fed a Western diet [122]. Higher numbers of α4β7-expressing CD4+ T cells were found in mice with steatohepatitis and a corresponding increase in the ligand MAdCAM-1 and antibody-induced blockade of α4β7 or MAdCAM-1 ameliorated steatohepatitis. Furthermore, vascular adhesion protein 1 (VAP1), an amino-oxidase expressed by LSECs, was involved in the recruitment of CD4+ T cells to the liver in NAFLD [123]. OX40 (CD134) is a costimulatory molecule of the TNF family expressed on T cells [124], and a recent study identified a role for OX40 in NAFLD [125]. Mice that received a high fat diet showed increased OX40 expression in CD4+ T cells and OX40 deficiency reduced CD4+ T cell infiltration, Th1 differentiation and alleviated steatohepatitis [125]. Furthermore, patients with NAFLD had increased serum levels of OX40 that correlated with disease severity. 

Th17 cells are a subset of CD4+ T cells marked by the production of IL-17 and are involved in numerous inflammatory disorders [126]. Th17 cells were increased in patients with NAFLD and improvement in NASH after bariatric surgery correlated with decreasing Th17 cells [127]. Similar findings were obtained in a mouse study using the high fat diet, where increased intrahepatic IL-17 was noted [128]. Furthermore, mice deficient in IL-17A, IL-17F or IL-17A receptor that received the MCD diet had increased steatosis but reduced steatohepatitis [129,130]. 

Regulatory T cells (Treg) are a subset of T cells with immunosuppressive capacity and an important contributor to liver tolerance [27]. Patients with NASH had lower frequencies of Treg both in blood and liver and conversely, the Th17/Treg ratio was increased [127]. Mechanistic studies showed that Treg in steatosis are more susceptible to apoptosis in response to oxidative stress [131]. A recent study showed that reducing Treg numbers in mice lacking CD80/CD86, aggravated steatohepatitis [132]. Loss of liver tolerance via Treg, mediated by Kupffer cells, was previously described in liver inflammation in mice and might provide a mechanistic link to the reduced frequencies of Treg in NASH patients [133]. Collectively, these studies indicate that effector functions of CD4+ and CD8+ T cells aggravate steatohepatitis and a disbalance of regulatory and inflammatory T cells contributes to steatohepatitis in humans and rodent models.

B cells accumulate in the livers of mice with steatohepatitis, secreting proinflammatory cytokines such as TNFα and IL-6 [134]. Furthermore, in mice with NASH, B cell-activating factor (BAFF), a cytokine critical for B cell survival and maturation, was upregulated in the liver [110]. In addition, serum BAFF levels were elevated in patients with NASH and correlated with severity of steatohepatitis and fibrosis [135]. A recent study investigated the effect of BAFF-deficiency in mice fed a high fat diet [136]. *Baff^-/-^* mice exhibited improved insulin sensitivity, reduced weight gain, hepatic steatosis and adipose tissue inflammation. This study was in line with a previous investigation noting ameliorated steatohepatitis in mice treated with a BAFF depleting antibody or in mice depleted in B cells [137]. Thus, the limited evidence available suggests a promoting role for B cells in steatohepatitis.

### 2.7. Involvement of Platelets

Platelets are the main cellular player in thrombosis and hemostasis, however, their role in inflammation and infection is increasingly appreciated. Platelets release antimicrobial molecules, influence immune responses and maintain vascular integrity during infection [138,139,140,141]. Furthermore, platelets play a key role in progression of obesity and metabolic syndrome, and activated platelets are known drivers of arteriosclerosis and cardiovascular risk in obesity [142,143]. A role for platelets has also been implicated in different liver diseases [144]. A cross-sectional study found an association of lower NAFLD prevalence in a US cohort with regular use of aspirin [145]. This was preceded by a study showing reduced inflammation and fibrosis with the use of antiplatelet agents in mice with steatohepatitis [146]. A recent study investigated the role of platelets more detailed in patients and mice with NASH [76]. The earliest changes observed, were platelet aggregates on Kupffer cells, preceding immune cell recruitment. Consequently, antiplatelet therapy reduced immune cell trafficking and mechanistically, platelet-derived GPIbα was found to be critical for NASH progression [76]. In a clinical trial, a slower NAFLD-to-NASH progression in patients on antiplatelet therapy was reported [76], thus offering a novel mechanistic avenue for NASH development and potential targets for chemoprevention of NASH-related complications. In addition to platelets, the plasmatic arm of the coagulation cascade was also shown to be involved in the pathogenesis of NAFLD. In mice fed a high fat diet, hepatic fibrin deposition preceded liver injury, thrombin levels in plasma were elevated and treatment with the thrombin inhibitor dabigatran reduced steatohepatitis and obesity in mice [147]. Together these studies hint at a potential interplay of coagulation and inflammation in NASH, however more mechanistic evidence is needed to understand the precise mechanism.

### 2.8. Hepatocyte Inflammatory Signaling

Besides their important role in metabolism, hepatocytes are involved in the innate immune response to infections or noninfectious injury by sensing endotoxins and secreting soluble inflammatory mediators [148]. In steatohepatitis, hepatocytes utilize pattern-recognition receptors to sense excess metabolites and increased endotoxin levels, and thus actively participate in local inflammation resulting in cell death [26]. Toll-like receptors (TLRs) expressed by hepatocytes are well studied and have a proinflammatory role in the NAFLD pathogenesis, including TLR2, TLR4, TLR5, and TLR9 [26]. Furthermore, intracellular inflammasome activation in hepatocytes in response to DAMPs, leads to the production of proinflammatory cytokines IL-1β and IL-18 [14]. In aggregate, activation of innate immune receptors in hepatocytes enhances downstream activation of intracellular adaptors and kinases including myeloid differentiation primary response 88, TIR-domain-containing adapter-inducing IFN-β, TNF-receptor-associated factors, TGF-β-activated kinase 1/JNK, and IκB kinases (IKKs) [26]. Furthermore, inflammatory signaling activates nuclear transcription factors including IRF, NF-κB and PPAR-α together propagating hepatic inflammation. The current body of evidence suggests that hepatocytes in NAFLD are not solely the target of necroinflammation, but also actively orchestrate and amplify immune responses. Interestingly, progression of NASH and transition to HCC might be underpinned by different hepatocyte signaling pathways. Oxidative stress in experimental obesity promoted STAT-1 and STAT-3 signaling in mice, leading to T cell infiltration, fibrosis and HCC [149]. Inhibiting STAT-1 signaling in hepatocytes prevented NASH development, but had no effect on HCC occurrence, and conversely, blocking STAT-3 prevented HCC without affecting NASH, potentially providing a mechanistic basis for HCC development independent of fibrosis and NASH with hepatocyte signaling critically involved.

## 3. Gut–Liver Axis

There is constant crosstalk between liver and intestinal microbiota, which is exemplified by the recent observation that microbial signals position Kupffer cells in their vascular location [150]. Our understanding of the gut microbiome has increased exponentially over the last decade and it is now viewed as an active contributor to physiology [151,152]. Furthermore, intestinal dysbiosis is associated with metabolic disease such as obesity [153,154,155,156,157], type 2 diabetes mellitus [158,159,160,161] and NAFLD [152,162,163,164]. Suggested mechanisms include bacteria or their products translocating to the liver through disrupted intestinal barrier, evoking liver inflammation via Toll-like receptors and inflammasome activation, aggravating steatohepatitis in the process (Figure 2). As evident from comparing inbred laboratory mouse strains to true “wild type” mice in natural forests, the composition of the gut microbiota shapes the responsiveness of hepatic immune cells, thereby contributing to the host’s immune fitness and resilience against diseases [165,166]. More recently, other components of the gut microbiome such as viroids and fungi were found to be dysregulated in NAFLD as well [167]. 

High fat diet induces milder steatosis in germ-free mice compared to conventionally housed mice [168,169]. Furthermore, the transmissibility of the NASH phenotype via gut microbiota was demonstrated [170]. In addition, a study identified a NAFLD-resistant and NAFLD-prone microbiota signature, and only germ-free mice on high fat diet reconstituted with the NAFLD-prone microbiota developed steatosis [171]. These findings were substantiated by two recent studies showing enhanced steatohepatitis in germ-free mice that received gut microbiota transplantation from NAFLD patients [172,173]. In patients, complexity of the microbiome was reduced [174]. A comprehensive review of microbiome signatures in NAFLD has recently been published [175]. Dietary cholesterol, which is a major lipotoxic molecule, was recently implicated in the development of NASH and HCC in mice by modulating intestinal microbiota [176]. Interestingly, mice fed a high fat/high cholesterol diet, showed characteristic dysbiosis along stages of NASH and HCC, which was mirrored in hypercholesteremic patients. Furthermore, anticholesterol treatment restored dysbiosis and prevented NASH and HCC, together suggesting a role for dietary cholesterol in mediating dysbiosis and subsequent NAFLD [176]. 

The intestinal barrier, which separates the host from the intestinal lumen, was shown to be dysfunctional in NAFLD leading to a leaky gut with increased gut–liver-crosstalk [177,178]. It is well established that patients with NAFLD have higher levels of systemic bacterial products such as lipopolysaccharide (LPS) [152]. It was shown experimentally that dietary components and obesity can directly perturb the intestinal barrier by changing tight junctions in the mucosa of the gut [99]. Furthermore, CX3CR1-expressing macrophages form a perivascular barrier in the intestine, which is disrupted in dysbiosis [179]. A recent study demonstrated that the presence of CX3CR1+ macrophages in the intestine had a protective effect in models of steatohepatitis and deficiency in CX3CR1 increased steatohepatitis by elevated endotoxin levels [180]. It remains elusive if leaky gut in NAFLD is a cause or consequence of disease [23], especially in humans where merely correlative studies exist to date. A recent study in mice concluded that intestinal barrier disruption was an early event in steatohepatitis [181]. Dysbiosis and gut vascular barrier disruption were found at the beginning of NASH, which was replicated by fecal microbiota transplantation of mice on high fat diet into control mice. Mechanistically, WNT/β-catenin signaling in endothelial cells was involved and importantly, upon genetic prevention of vascular barrier disruption, NASH development was attenuated [181]. Furthermore, another study found that in experimental NASH using the high fat diet, inducing concomitant colitis aggravated steatohepatitis, increased endotoxins, TLR4, TLR9, and liver fibrosis [182]. Similarly, in mice deficient in junctional adhesion molecule A (JAM-A), which models leaky gut, high fat diet led to more severe steatohepatitis [183]. These studies clearly establish the gut microbiome, leaky gut and microbial translocation as aggravating factors in steatohepatitis. Of note, patients often have a number of metabolic comorbidities making the liver related effect difficult to disentangle, which has to be taken into consideration when deciphering the role of intestinal microbes further [14]. 

Importantly, the gut–liver axis is not unidirectional (metabolic signals from the gut reach the liver via the portal vein), but also involves signals from the liver impacting the gut. The gut–liver axis is partially mediated by bile acids, which are synthesized and secreted in the liver. Bile acids are involved in the intestinal uptake of lipids, are transported back to the liver via the enterohepatic circulation and act on farnesoid X receptor (FXR), expressed by hepatocytes to regulate glucose and lipid metabolism [108]. Bile acids and the intestinal microbiome have a reciprocal relationship, as bile acids shape microbiome composition through direct effects and via FXR-induced production of antimicrobial peptides, and the gut microbiome modifies the bile acid pool through enzymatic activities [184]. Dysbiosis can alter the pool of bile acids, and on the other hand altered bile acid composition can lead to dysbiosis [14]. A recent study demonstrated that gut microbiota promoted obesity and liver steatosis in mice via FXR signaling [185]. Along those lines, another study in patients attributed the positive effects of bariatric surgery on NASH to FXR signaling [186]. Of note, an increase in conjugated bile acids in patients with NASH vs. steatosis was identified and conjugated bile acids correlated with disease severity [187]. However, further mechanistic studies are warranted to investigate the potential mechanisms of altered bile composition in NASH [188]. Importantly, two landmark studies have recently identified that the balance between proinflammatory Th17 and anti-inflammatory Treg differentiation of CD4+ T cells is mediated by bile acids in the intestine and it is intriguing to speculate, whether the Th17-Treg disbalance featured in NALFD, might be driven by altered bile acids in the intestine with dire consequences to the liver [189,190].

## 4. Adipose Tissue–Liver Crosstalk

NAFLD is part of a spectrum of metabolic disorders and thus has been viewed as a multisystem disease [191,192,193]. Recent studies have uncovered important aspects of how the adipose tissue and adipose inflammation contribute to steatohepatitis [194] (Figure 2). Excess nutrients lead to the accumulation of fat and hypertrophy of adipose tissue. This begets an immune response with recruitment of proinflammatory cells. Excessive production of proinflammatory cytokines by adipose tissue macrophages is considered critical in conveying a systemic inflammatory environment [195]. It was shown that the number of macrophages in adipose tissue correlated with metabolic syndrome exacerbation [196]. Gut-derived LPS induced inflammatory pathways in adipose tissue through TLR4 signaling, enhancing the recruitment of proinflammatory monocytes and TLR4 deficiency alleviated inflammation [197,198]. Consequently, inhibiting the CCL2-CCR2 axis improved glucose tolerance in diet-induced obesity [199,200,201]. Interestingly, a sophisticated study using single-cell sequencing in humans and mice, recently identified a macrophage subset with high expression of TREM-2 emerging during obesity [202]. Mechanistically, TREM-2 was a key regulator to limit adipocyte hypertrophy and subsequent hyperlipidemia and insulin resistance. The crosstalk between adipose tissue and liver in steatohepatitis is in part mediated by adipokines, and in principle, leptin aggravates NASH, whereas adiponectin is viewed to have beneficial effects (Figure 2) [203]. In mice with steatohepatitis, leptin induced Kupffer cell activation via iNOS and NADPH oxidase, and subsequently increased sensitivity towards low-dose LPS, ultimately corroborating liver inflammation [204,205]. Furthermore, fatty-acid binding protein 4 (FABP4), which is produced by adipocytes and macrophages, was elevated in NASH patients, and served as a biomarker to identify progression in NAFLD [206,207]. FABP4 induces intracellular lipid accumulation [208], and was shown to be associated with increase of certain triglycerides in mice with steatohepatitis [209]. The principle association between adipose tissue inflammation and NASH was also established in patients, as inflammatory markers in adipose tissue correlated with progression of NASH [210]. 

Obesity was also implicated to influence bone marrow production of inflammatory progenitor cells. In mouse models of obesity, a study found monocytosis, neutrophilia and expansion of myeloid progenitors in bone marrow [211]. Mechanistically, adipose tissue derived S100A8/A9 stimulated IL-1β release in adipose tissue macrophages, which in turn promoted increased myeloid precursor production via the IL-1 receptor. Another current study documented an LPS-induced TLR-4-dependent shift in obese mice in their bone morrow gene signature towards enhanced myeloid and suppressed lymphoid genes, potentially explaining the predominant myeloid infiltrate in adipose tissue [212]. In addition, a distinctive inflammatory gene signature in bone marrow myeloid precursors during experimental steatohepatitis was identified using single-cell RNA sequencing, mirroring liver myeloid cells [213].

## 5. The Transition to NAFLD-Associated HCC

NASH is marked by necroinflammation and lipid accumulation, however, how this microenvironment of aberrant metabolism, low grade inflammation and ongoing liver regeneration contributes to DNA damage and cancer is still poorly understood [23]. In principle, chronic inflammation and hepatocyte death induce liver regeneration and hepatocyte proliferation, leading to mutated cells, fostering an environment that favors hepatocarcinogenesis [214]. Mechanistic studies of the NASH-to-HCC transition are challenging due to the fact that diet-induced models either fail to induce HCC or require long treatment duration [215]. Therefore, an additional genetic modification or carcinogen application are utilized in rodent models, making the findings harder to translate. 

### 5.1. The Role of the Adaptive Immune System

The adaptive immune system is critically involved in tumor surveillance, but also promotes the underlying steatohepatitis, it is therefore important to understand the role of T and B cells in the progression from NASH to HCC. In a mouse study using long-term choline-deficient high fat diet (CD-HFD), a NASH-induced HCC transition was established [117], and CD8+ T cells and NKT cells were identified as the main drivers of inflammation, and activation of hepatocellular lymphotoxin-β receptor (LTBR) and canonical NF-kB signaling promoted NASH-to-HCC transition [117]. Moreover, lack of CD8+ T cells and NKT cells reduced tumor development. Furthermore, in the same model it was shown that platelets were involved in the NASH-to-HCC transition, and antiplatelet therapy was successful in reducing tumor development [76]. However, if platelets play a mechanistic role in HCC development was not assessed. While platelets might aid CD8+ T cells early in driving steatohepatitis, they might dampen tumor growth in advanced stages of NASH via recruitment of tumor-fighting T cells, however, this warrants further investigation [216]. In contrast, in a different study using the mouse model of urokinase plasminogen activator-overexpressing mice fed a high fat diet, ablation of CD8+ cells promoted HCC development and IgA+ expressing plasma cells were seen as immune suppressing cells found in abundance in mice and NASH patients, expressing PD-L1 and IL-10 [217]. This is in line with the general conception, that CD8 T cells exert antitumor functions, and a larger number of tumor-infiltrating CD8+ T cells has been shown to correlate with improved survival in HCC [218]. Furthermore, CD8+ T cells were shown to decrease with advanced disease and their ability to produce IFN-γ was impaired [219], indicating they might lose antitumor function over time. In line with this, experimental studies with tumor cell implantation in mice suggested that steatohepatitis impaired T-cell-directed immune-oncological therapies against tumors [220].

A recent study found that selective loss of CD4+ T cells accelerated HCC growth in a model of NASH in mice with hepatocyte-specific overexpression of MYC [221]. CD4+ T cell loss was due to mitochondrial oxidative stress, resulting from dysregulated lipid metabolism. Linoleic acid caused oxidative damage and blocking ROS restored CD4+ T cells and delayed HCC growth. Studies in patients corroborated a tumor enhancing function of Treg, as Treg numbers infiltrating HCCs correlate with poor outcome [222,223]. A recent study identified an interesting feed forward loop, by which Th17 cells promote NASH and HCC in mice [224]. Mice overexpressing the transcriptional repressor unconventional prefoldin RPB5 interactor (URI) were fed a high fat diet; it was demonstrated that DNA damage triggered IL-17A release, which promoted neutrophil infiltration in adipose tissue, causing insulin resistance, steatohepatitis and HCC, which was prevented by blocking IL-17. The discrepancies regarding the roles of T cells in the NASH-to-HCC transition are intriguing and it is important to mention the predicament that the immune system faces in the context of steatohepatitis: on the one hand T cell driven inflammation contributes to aggravating steatohepatitis but T cells also exert important antitumor functions, that might be overridden in tumor development. 

### 5.2. The Role of the Innate Immune System

The role of innate immunity, particularly of myeloid cells like Kupffer cells, monocytes and macrophages, in the NASH-to-HCC transition is ambiguous. In HCC, blocking the CCL2-CCR2 axis appears to be protective and CCL2 was overexpressed in human HCC and associated with adverse prognosis [225]. An interesting immunological commonality between NASH and HCC might be rooted in inflammasome activation in liver macrophages. Robust evidence has shown that NLRP3 inflammasome activation in liver macrophages leads to hepatocyte death and lack of NLRP3 inflammasome attenuates inflammation and fibrosis in NASH [226,227]. Inflammasome activation provides a link to HCC via downstream production of IL-6, which was shown to promote HCC in mice [228,229]. Furthermore, reactive oxygen species produced by macrophages contribute to NASH and HCC [230,231]. In a mouse model of toxin induced HCC, Kupffer cell TREM-1 was mechanistically involved in HCC development [232]. Single-cell RNA sequencing studies from human (not NAFLD-related) HCC revealed a remarkable heterogeneity of tumor-associated macrophages, indicating the presence of tumor-promoting (e.g., immune-suppressive myeloid derived suppressor cells) and antitumor (e.g., antigen-presenting, immunogenic) populations [233]. The liver in NASH is populated with an abundance of macrophages of different phenotypes on the spectrum of proinflammatory and repair. It is conceivable that repair macrophages might inadvertently favor tumor development, but more mechanistic evidence is needed. 

NK cells have a crucial role in tumor surveillance and a breakdown of that function favoring HCC development was proposed, based on the observation that NK cells of patients with HCC were inhibited by myeloid-derived suppressor cells [234]. A role for CXCR6-expressing NKT cell removal of senescent hepatocytes and thus their role in HCC was also established [235]. CXCR6-deficiency led to higher tumor burden in mice and lower numbers of iNKT cells and CD4+ T cells. Furthermore, patients with HCC had lower peritumoral CXCR6 expressing lymphocytes compared to cirrhosis, which taken together suggests a tumor surveillance function of NKT cells in the liver. Another study found that expression of the ligand of CXCR6, CXCL16, by LSEC regulated NKT cell recruitment in models of liver tumor and metastasis and NKT cells had an antitumor effect via IFN-γ [236]. Interestingly, CXCL16 expression was regulated by microbiome dependent primary-to-secondary bile acid conversion, thus implicating immune surveillance of the microbiome via bile acids as mediators. 

Neutrophil-induced oxidative stress was shown to diminish the capacity to repair DNA damage, thus contributing to HCC [26]. Along those lines, tumor infiltrating CCL2- and CCL17-expressing neutrophils mediated immune evasion by promoting anti-inflammatory macrophages and Treg [237]. In patients, these neutrophils correlated with poor prognosis, and in mice, tumor-associated neutrophils conferred resistance to sorafenib treatment and their depletion enhanced tumor control [237]. Furthermore, a recent study investigating the role of neutrophil extracellular traps in NAFLD and associated HCC demonstrated that blocking the formation of NETs, was able to diminish tumor growth in mice [75]. 

Bile acids might also play a role in the transition from NASH to HCC and an association of altered bile metabolism and HCC has been suggested [238]. Experimentally, it was shown that high levels of bile acids can cause DNA damage in hepatocytes, promoting carcinogenesis [239]. In an experimental study, FXR-deficient mice developed spontaneous liver tumors and tumor development was prevented by activating intestinal FXR signaling [240,241].

## 6. Therapeutic Implications

In light of the multifaceted contributions of the immune system in NAFLD, targeting inflammatory pathways is a promising avenue. However, on the road from steatosis to NASH and HCC, the immune system might have opposing functions: as proinflammatory drivers of steatohepatitis, and during more advanced stages of disease, by nurturing an anti-inflammatory and regenerative milieu, jointly favoring tumor development. Furthermore, cancers are known to orchestrate an intrinsic inflammatory microenvironment [242]. As such, mouse models have shown that liver tumors driven by the same oncogenes can have different transcriptomes depending on the inflammatory environment [214]. With this in mind, successful therapeutic intervention hinges on the specificity of the targeted cell/molecule and likely the timepoint of intervention. 

Targeting proinflammatory monocytes via the dual CCR2/CCR5 antagonist cenicriviroc showed promising results in mouse studies [64]. Furthermore, in the phase 2b clinical CENTAUR trial, reducing fibrosis stage ≥ 1 without worsening of NASH was achieved in more patients on cenicriviroc compared to placebo [243]. Cenicriviroc has been evaluated in a phase 3 study (AURORA; NCT03028740), but this trial was terminated because fibrosis regression was not achieved upon interim analysis. In the phase 2 TANDEM trial (NCT03517540), cenicriviroc is tested in combination with the FXR agonist tropifexor. Another FXR agonist, obeticholic acid, showed promising results in an interim analysis of a phase 3 trial with significant improvement in fibrosis [244]. Peroxisome-proliferator-activated receptors (PPARs) comprise a family of receptors with numerous effects on lipid and glucose metabolism and furthermore are expressed by macrophages mediating anti-inflammatory effects [245]. The pan-PPAR agonist lanifibranor showed promising results in mouse models of steatohepatitis, reducing fibrosis and inflammation [246], and furthermore, results of a phase 2 clinical trial were encouraging, as steatohepatitis and fibrosis were diminished compared to placebo [245]. Multiple PPAR agonists are currently tested in clinical trials and results are pending. It is worth mentioning that FXR and PPAR agonists act on hepatocytes and adipocytes besides the documented anti-inflammatory effects, which needs to be accounted for when studying their effects on steatohepatitis.

Since the gut microbiome might be involved in aggravating steatohepatitis, modifying intestinal microbes, e.g., by fecal microbiota transplantation (FMT), could be a therapeutic pillar in NAFLD. In mice, FMT delivered promising results, alleviating steatohepatitis in recipients [247,248], and clinical trials assessing FMT in humans are underway (NCT04465032, NCT02469272, NCT02721264). A recently completed small trial failed to show improved insulin resistance after FMT, however reported improved intestinal permeability [249]. Changes in gut microbiome might also be useful as a prognostic or diagnostic tool. Two recent studies provided initial evidence using an integrated approach with data from metagenomics, metabolomics and transcriptomics as biomarkers to correlate with liver fibrosis in obese patients [250,251].

Increased understanding of cells of the adaptive immune system might extend therapeutic options in NAFLD. Some studies suggested a role for B cells in the pathogenesis of NASH and BAFF was shown to be elevated [110,136,137], thus targeting BAFF with the monoclonal antibody belimumab might be feasible. A different route is targeting regulatory T cells via the T cell receptor-associated molecule CD3, an approach which was shown to ameliorate atherosclerosis in mice using the monoclonal antibody OKT3 [252]. Interestingly, in obese mice treated with OKT3, liver steatosis and hepatocyte death decreased and in a phase IIa clinical trial in patients with biopsy-proven NASH and type 2 diabetes mellitus, OKT3 improved aminotransferase levels and insulin resistance [253,254,255]. 

Squalene epoxidase, an enzyme involved in cholesterol biosynthesis, was recently identified as an oncogene driving NAFLD-associated HCC development via regulation of cholesteryl esters and reactive oxygen species [256]. Interestingly, targeting squalene epoxidase with an approved antifungal drug, terbinafine, suppressed NAFLD-induced HCC growth in vivo and in vitro, highlighting the therapeutic potential of terbinafine in the context of NAFLD-induced HCC.

In recent years, immune checkpoint blockade has been introduced in the treatment of HCC [257]. Checkpoint inhibitors aim to restore immune control of tumors by disrupting coinhibitory receptors such as PD-1 or CTLA-4, thus enhancing T cell responses. This has led to the approval of atezolizumab (anti-PD-L1 antibody) and bevacizumab (anti-VEGF) for patients with unresectable HCC [258]. It is noteworthy, that despite the successful IMBRAVE-150 trial, overall survival in anti-PD-L1 treated patients was low, despite HCC being an inflammation-driven cancer [228]. A novel approach might be to enhance the immunogenicity of HCC by combining checkpoint inhibitors with chemoembolization [259]. HCC in the context of NASH is especially challenging from an immunotherapeutic angle, because “unchecking” the immune system to fight HCC might aggravate the underlying NASH. Furthermore, while checkpoint inhibitors are aimed at restoring adaptive immunity, effects of the innate arm of the immune system have yet to be tackled, but a combination therapy might offer promise. 

## 7. Conclusions

NAFLD with associated HCC represents a challenging systemic disease with numerous mechanisms, such as metabolic disturbance, lipotoxicity, adipose tissue inflammation, and dysbiosis, contributing to liver inflammation. The immune system plays the role of a double agent, contributing to progression from steatosis to NASH, yet also allowing hepatocarcinogenesis due to insufficient tumor surveillance. Various immune cells seem to change their phenotype during disease course. Furthermore, a hallmark of metabolic inflammation are fluctuations between flares of inflammation and resolution, leaving the immune system at a disbalance between inflammatory and repair mechanisms. Novel methodologies have greatly increased our understanding of the heterogeneity of immune cells involved and unravelling the spaciotemporal kinetics and interplay of these populations offers new therapeutic avenues. From a basic science perspective, descriptive single-cell and microbiome work warrants future functional studies, to fully elucidate the complex interplay of immunity, metabolism and microbiome. Clinically, more personalized diagnostics based on microbiome, tumor immune landscape and liver immune cell heterogeneity, will help to identify and tailor therapies for different patient cohorts. 

## Figures and Tables

**Figure 1 cancers-13-00730-f001:**
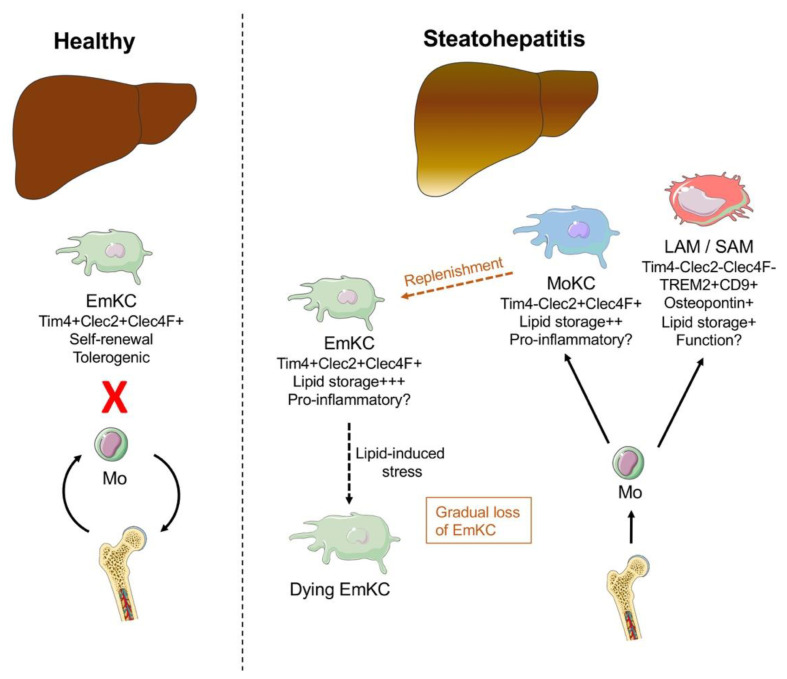
The fate of monocytes and macrophages in steatohepatitis. In homeostasis (left panel), the liver is populated with embryonic Kupffer cells (EmKC) that are maintained by self-renewal and have, in principle, a tolerogenic phenotype. Specific markers for Kupffer cells in mice are Tim4, Clec4F and Clec2. In steatohepatitis (right panel), lipid-induced stress impairs embryonic Kupffer cell replication, inducing cell death and resulting in their gradual loss. Infiltrating monocytes have at least two distinct fates: as monocyte-derived KC (MoKC) with largely overlapping gene signatures albeit more proinflammatory. They occupy the Kupffer cell niche and replenish EmKC. Alternatively, monocytes give rise to lipid-associated macrophages (LAM) or scar-associated macrophages (SAM) with the expression of CD9, TREM2 and osteopontin, show differences in lipid and inflammatory genes and a location in proximity to the fibrotic niche.

**Figure 2 cancers-13-00730-f002:**
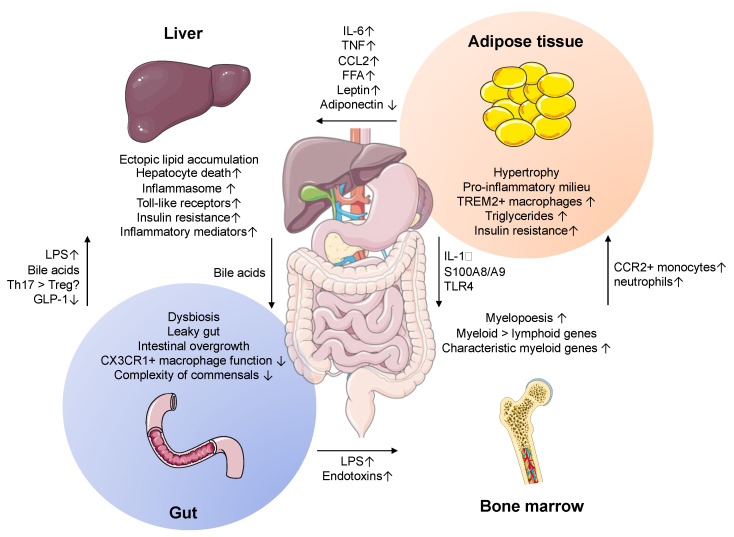
Multiple systems drive inflammation and progression of nonalcoholic fatty liver disease (NAFLD). An interorgan crosstalk between adipose tissue, intestine, bone marrow, and liver triggers the inflammatory cascade in steatohepatitis. In adipose tissue (top right), excess lipids lead to hypertrophy and induce the release of proinflammatory mediators such as IL-1β, IL-6, TNF, and CCL2 and influx of inflammatory cells. Furthermore, disturbed adipokines (low adiponectin and high leptin) and increased free fatty acids (FFA) induce lipotoxic effects in the liver. In bone marrow (bottom right), inflammatory mediators increase myelopoiesis and decrease lymphopoiesis. In the gut (bottom left), disrupted intestinal barrier leads to increased levels of pathogen-associated molecular patterns (PAMPs), such as LPS that enter the circulation and portal vein, stimulating inflammation in liver and myelopoiesis in the bone marrow. Increased systemic inflammation, hepatocyte death and endotoxins in concert drive liver inflammation. In a vicious cycle, the liver produces more inflammatory mediators such as acute-phase proteins and complement, further corroborating inflammation. Bile acids from the liver modify the intestinal microbiota and regulate inflammatory/regulatory cells, including T cells.

**Table 1 cancers-13-00730-t001:** Immune cell populations and their role in steatohepatitis.

Population	Marker		Role in NAFLD
	Humans	Mice	
Monocytes/Macrophages	CD14++CD16- CD14+CD16+ CD14-CD16++ CD68	Ly6C^hi^CCR2+ Ly6Cl^ow^CX3CR1+ F4/80	Proinflammatory role in aggravating NASH, hepatocyte damage and fibrosis (CCR2+) Repair function/healing (CX3CR1+ patrolling monocytes) Replenishment of embryonic Kupffer cells CD9+TREM-2+ scar/lipid-associated macrophages (function?)
Kupffer cells	CD68 CRIg	Clec4F Tim4 F4/80	Detection of DAMPs, PAMPs Proinflammatory cytokine release Impaired renewal and increased cell death
Neutrophils	CD15 CD66b CD16	Ly6G Gr1	Promote NASH via the release of effector molecules (proteases, elastase, myeloperoxidase, ROS) NETs increased in patients, blocking NETs beneficial in mice MPO increased in human NASH, mice lacking MPO or neutrophil depletion protected from NASH
DCs	CD1c CD83 CD141 CD123 CD303/CD304	CD103 CD11c CD11b CD205 CD317	Bridging function between innate and adaptive immunity CD11c-dependent depletion exacerbated steatohepatitis CD103+ cDCs alleviate steatohepatitis in mice CX3CR1+ myeloid DCs aggravate NASH in mice
NK cells	CD56 CD244	CD49b NK1.1 (CD161) NKp46	Antifibrotic activity by targeting hepatic stellate cells via receptors NKG2D, NKp46
NKT cells	CD3 CD56 Vα24	CD3 NK1.1 (CD161) CXCR6	Increased in human NASH and cirrhosis CXCR6+ iNKT cells aggravate steatohepatitis Reduced in early stages and skewed towards Th1 profile
T cells	CD3 CD4 CD8 FoxP3 (Treg)	CD3 CD4 CD8 FoxP3 (Treg)	CD8+ T cells promote NASH by aggravating injury IFN-producing CD4+ T cells increased in patients with NASH, promote NASH in mice Th17 cells are increased in patients and mice, Th17 cells promote NASH in mice Treg cells are decreased in patients and mice
B cells	CD19	CD19	BAFF elevated in serum of patients with NAFLD Blocking BAFF in mice ameliorated steatohepatitis TNF-α and IL-6 producing B cells found in mice with steatohepatitis suggesting a promoting role

Abbreviations: Ly6C, lymphocyte antigen 6 complex; CCR2, C-C chemokine receptor type 2; CX3CR1, fractalkine receptor; TREM2, triggering receptor expressed on myeloid cells 2; CRIg, complement receptor of the immunoglobulin family; Clec4F, C-type lectin domain family 4 member F; Tim4, T-cell membrane protein 4; DAMPs, damage-associated molecular patterns; PAMPs, pathogen-associated molecular patterns; ROS, reactive oxygen species; NETs, neutrophil extracellular traps; DCs, dendritic cells; NK cells, natural killer cells; NKT cells, natural killer T cells; CXCR6, C-X-C chemokine 6; Th1, T helper cells 1; IFN, interferon; FoxP3, forkhead box P3; Treg, regulatory T cells; BAFF, B-cell activating factor; TNF; tumor necrosis factor.

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
