# Peer review of "Inflammatory Mechanisms Underlying Nonalcoholic Steatohepatitis and the Transition to Hepatocellular Carcinoma"

_cancers, 2021, doi:10.3390/cancers13040730_

Round 1

Reviewer 1 Report

Non-alcoholic fatty liver disease (NAFLD) is a disease of increasing interest as its prevalence is on the rise. The number of NAFLD induced HCC has increased dramatically due to the rising incidence of obesity and metabolic syndrome. In this review article, Peiseler et al, have discussed the molecular mechanisms of NAFLD-induced HCC with special focus on inflammation in multiple perspectives. Firstly, they discussed the pivotal role of inflammation in NAFLD-induced HCC, with the involvement of macrophages, neutrophils, dendritic cells, natural killer cells, natural killer T cells, and platelets. Next, they discussed the involvement of adaptative immune cells, particularly in various subsets of T cell populations. Increasing attention has been gained linking liver and intestinal microbiota, which has been exemplified by the recent observation of microbiome signaling in positioning of kuffer cells. They have also discussed in depth about the gut-liver axis, which links gut microbiota, inflammation and T cells with NAFLD development. Next, they have discussed the role of adipocytes in development of NAFLD disease. They have further provided several important mechanisms for NAFLD-induced HCC including inflammasome and immune dysfunction involving CD4+T cells, neutrophils and  NK cells. Lastly, they provided the potential therapeutic strategies for treatment of NAFLD-induced HCC.  All in all, this review provides a novel mechanistic insight for NAFLD-induced HCC, in which NAFLD is a rising risk factor for the development of HCC. This review is comprehensive, up-to-date and with in-depth discussions, which will be appreciated not only by gastrohepatologists but also by oncologists and others in the field of cell biology. Please find the comments below for further improvement of this review article.

  1. Please include the discussion of FABP4 in development of adipocyte tissue-liver cross-talk, as FABP4 is the main adipocyte marker.
  2. For the therapeutic implications, please also include targeting squalene epoxidase for NAFLD-induced HCC by Terbinafine (Liu et al., Sci Transl Med, 2018).
  3. For the gut-microbes axis, please also include the paper on the role of dietary cholesterol in NAFLD-induced HCC (Zhang et al., Gut, 2020).

Author Response

We would like to thank the reviewer for the thorough assessment of our manuscript and the helpful additions that greatly enhance the scientific value of our review. Please find attached a point-by-point response to the issues raised:

  • Please include the discussion of FABP4 in development of adipocyte tissue-liver cross-talk, as FABP4 is the main adipocyte marker.

We added a paragraph on the role of FABP4 (ll. 542 – 546).

  • For the therapeutic implications, please also include targeting squalene epoxidase for NAFLD-induced HCC by Terbinafine (Liu et al., Sci Transl Med, 2018).

We added this interesting reference (ll. 731 - 736) along with a sentence summarizing the study.

  • For the gut-microbes axis, please also include the paper on the role of dietary cholesterol in NAFLD-induced HCC (Zhang et al., Gut, 2020).

The suggested reference along with a paragraph was added (ll. 454 – 460)

Reviewer 2 Report

In this well written review, Peiseler M and Tacke F. summarize immune-mediated mechanisms of NASH and the role that immune-system play in the transition to associated HCC. In a complex multisystem disease, such as NAFLD, the immune system is constantly involved between inflammation, repair and tumor  surveillance - in light of novel immunotherapies for liver cancer, an improved understanding of the precise immune pathways and immune cells description is necessary and the authors in this review paint a precise and fully covered image of the immune system contribution to the transition to HCC. I support the publication of this review with minor points.

  • Line 51 and 54. Consistency on % format 1-2% or 1 – 2%.
  • Line 82. Could be interesting mention more on genetic (if any) background that could create a susceptible liver for NASH. Is it an aspect to be carefully mentioned.
  • What are Tim4, Clec2 and Clec4F mentioned in Figure 1? Not explained in the legend and in the text.
  • Line 143. (Fig. 1) should be removed. The statement before it does not represent what Fig. 1 is showing.
  • In Figure 2 IL-1α correct, there is a square instead α.

Author Response

We would like to thank the reviewer for carefully and thoroughly assessing our manuscript and for the helpful comments we received. Please find below a detailed point-by-point response to the comments:

  • Line 51 and 54. Consistency on % format 1-2% or 1 – 2%.

Revised as suggested.

  • Line 82. Could be interesting mention more on genetic (if any) background that could create a susceptible liver for NASH. Is it an aspect to be carefully mentioned.

We added three sentences on genetic modifiers in NAFLD and cited a comprehensive review on this topic for further reading as we have already extended the word limit (ll. 84 – 90).

  • What are Tim4, Clec2 and Clec4F mentioned in Figure 1? Not explained in the legend and in the text.

We added this important information to the figure legend of Fig. 1.

  • Line 143. (Fig. 1) should be removed. The statement before it does not represent what Fig. 1 is showing.

This sentence was deleted as requested.

  • In Figure 2 IL-1α correct, there is a square instead α.

We are afraid that there seems to have been a technical issue in the transfer, as the figures we uploaded as well as the figure in the manuscript we received back shows “IL-1b“, as intended. Hopefully this will be resolved before publication.

Reviewer 3 Report

The present Review by Peiseler and Tacke deals with a very extensive argument, i.e. the inflammatory mechanisms involved in NAFLD-NASH-HCC progression, developing the role of the single cell types, as well as the role of liver-gut axis and lipid-liver crosstalk. 

The Review is comprehensive, with adequate and updated references; English language seems correct to me (albeit English is not my mother tongue). 

I have just some minor corrections to suggest:

  1. The whole part 2 ("the role of inflammation in NAFLD") is very wide and comprehensive, with maybe too much information to be assimilated in one read: I suggest to add a table/figure summarizing the main inflammatory cells described, with the corresponding main activities and roles in NAFLD/NASH.
  2. In the Introduction, pag 2 raws 47-51, the reported percentages (15-25% of individuals...another 25%...representing 1-2% of all adults) are not clear. Please rephrase the sentence.

Author Response

We thank the reviewer for the helpful and thorough assessment of our manuscript. We fully agree with the proposed changes. Please find below a detailed response to the comments:

  • The whole part 2 ("the role of inflammation in NAFLD") is very wide and comprehensive, with maybe too much information to be assimilated in one read: I suggest to add a table/figure summarizing the main inflammatory cells described, with the corresponding main activities and roles in NAFLD/NASH.

We fully agree with the reviewer’s assessment and have now included a summarizing table replacing Figure 3. In table 1 we list the discussed immune cell populations, their characteristic markers in humans and mice, as well as a summary of their main function in NASH.

  • In the Introduction, pag 2 raws 47-51, the reported percentages (15-25% of individuals...another 25%...representing 1-2% of all adults) are not clear. Please rephrase the sentence.

We have rephrased this sentence as requested and hope it is clear now.

Reviewer 4 Report

Hepatic inflammation is a pivotal part of NAFLD/NASH as well as HCC. This review provides up-to-date and comprehensive image on how various immune cells contributes to liver disease progression, which arises the interest of potential readers. Few comments are;

Major points

- Reorganization of some sections is highly recommended: As per this reviewer’s recommendations, [Section 2.1] needs to be organized more readily. [Section 5] needs to be separated into two or three sub-sections.

- Some sections are too descriptive, just simply listing the previous findings, which it may be difficult for the readers to follow up, unless detail explanations are provided. E.g., Line 280-300 on CD8 and CD4 T cells. It is recommended to carefully provide summary statement in each section, which can help the readers have a clear image of specific immune cell functions in liver diseases.

- It is better to have a summary table of immune cell surface markers, which are described in Section 2.

- Please include few previous studies: [Section 2.7] Involvement of platelets – this can be further expanded into thrombin and tissue fibrin accumulation as reported by Dr. Luyendyk’s group. Also, as published in 2018, inflammatory signaling in hepatocytes (STAT1 and STAT3 signaling) plays distinct roles in NASH and HCC (PMID: 30454647). Related to this paper, is it possible to have a separate paragraph/section of hepatocyte inflammatory signaling (not immune cells)?

- Therapeutic implication of few nuclear receptor agonists (FXR and PPARs, lines 580-590) should be carefully mentioned since their primary targets are not an inflammatory signaling, although anti-inflammatory effect may be partially involved in. These agonists primarily affect hepatocytes and adipocytes (as well as immune cells) to show beneficial effects.

Minor points

- Check abbreviations. Some are not necessary (DAMPs, HRG, GILZ etc). Some should be spelled out (URI in line 500).

- Line 128-130. It is understandable M2 macrophage is also a part of the story. However, without any introduction on anti-inflammatory M2 macrophage, it is better to remove or separate into new paragraph (only describes anti-inflammatory Kupffer cells).

- Please avoid using some repetitive expressions like “the authors”.

- Check spelling; IL-1beta in Figure 2.

- It is recommended to redesign Figure 3, although it is not essential.

- No reference in line 174.

Author Response

We thank the reviewer for the thorough assessment of our manuscript and the helpful comments which we feel greatly improve the quality of our work. Please find below a detailed point-by-point reply to the issues raised:

Major points

  • Reorganization of some sections is highly recommended: As per this reviewer’s recommendations, [Section 2.1] needs to be organized more readily. [Section 5] needs to be separated into two or three sub-sections.

We thank the reviewer for this comment. We reorganized section 2.1. and subdivided section 5. We hope this helps to enhance the readability of the manuscript.

  • Some sections are too descriptive, just simply listing the previous findings, which it may be difficult for the readers to follow up, unless detail explanations are provided. E.g., Line 280-300 on CD8 and CD4 T cells. It is recommended to carefully provide summary statement in each section, which can help the readers have a clear image of specific immune cell functions in liver diseases.

We agree with this assessment and have now added summary statements following individual sections (e.g. ll 289-290 or ll. 362 – 364).

  • It is better to have a summary table of immune cell surface markers, which are described in Section 2.

We fully agree with the reviewer’s take and have now included a summarizing table replacing Figure 3. In table 1 we list the discussed immune cell populations, their characteristic markers in humans and mice, as well as a summary of their main function in NASH.

  • Please include few previous studies: [Section 2.7] Involvement of platelets – this can be further expanded into thrombin and tissue fibrin accumulation as reported by Dr. Luyendyk’s group. Also, as published in 2018, inflammatory signaling in hepatocytes (STAT1 and STAT3 signaling) plays distinct roles in NASH and HCC (PMID: 30454647). Related to this paper, is it possible to have a separate paragraph/section of hepatocyte inflammatory signaling (not immune cells)?

We are thankful for this important addition to our paragraph on non-conventional immune cells and added Dr. Luyendyk’s work (ll. 397 – 403) together with a paragraph headlined “hepatocyte inflammatory signaling” detailing the study by Grohmann et al.

  • Therapeutic implication of few nuclear receptor agonists (FXR and PPARs, lines 580-590) should be carefully mentioned since their primary targets are not an inflammatory signaling, although anti-inflammatory effect may be partially involved in. These agonists primarily affect hepatocytes and adipocytes (as well as immune cells) to show beneficial effects.

We added this information.

Minor points

  • Check abbreviations. Some are not necessary (DAMPs, HRG, GILZ etc). Some should be spelled out (URI in line 500).

We have made the requested changes.

  • Line 128-130. It is understandable M2 macrophage is also a part of the story. However, without any introduction on anti-inflammatory M2 macrophage, it is better to remove or separate into new paragraph (only describes anti-inflammatory Kupffer cells).

The sentence was removed.

  • Please avoid using some repetitive expressions like “the authors”.

We revised the manuscript accordingly and deleted those repetitions.

  • Check spelling; IL-1beta in Figure 2.

  • There seems to have been a technical issue as the figures we uploaded as well as the figure in the manuscript we received back shows “IL-1b“, as intended. Hopefully this will not occur during publication.

  • It is recommended to redesign Figure 3, although it is not essential.

We have removed Fig. 3 as it was misleading and oversimplified and have replaced it with a table summarizing immune cell populations in NASH.

  • No reference in line 174.

A reference was added.